# Visual Explanations of Differentiable Greedy Model Predictions on the Influence Maximization Problem

Mario Michelessa [1], Christophe Hurter [2], Brian Y. Lim [1], Jamie Ng Suat Ling [3], Bogdan Cautis [4] and Carol Anne Hargreaves [1,*]

1 Department of Statistics and Data Science, Faculty of Science, National University of Singapore, Singapore 117546, Singapore; mario.michelessa@u.nus.edu (M.M.); brianlim@comp.nus.edu.sg (B.Y.L.)
2 ENAC, Université de Toulouse, 31400 Toulouse, France; christophe.hurter@enac.fr
3 Institute for Infocomm Research, A*STAR, Singapore 138632, Singapore; jamie@i2r.a-star.edu.sg
4 Department of Computer Science, University of Paris-Sud, 91405 Orsay, France; bogdan.cautis@u-psud.fr
* Correspondence: carol.hargreaves@nus.edu.sg

**Abstract:** Social networks have become important objects of study in recent years. Social media marketing has, for example, greatly benefited from the vast literature developed in the past two decades. The study of social networks has taken advantage of recent advances in machine learning to process these immense amounts of data. Automatic emotional labeling of content on social media has, for example, been made possible by the recent progress in natural language processing. In this work, we are interested in the influence maximization problem, which consists of finding the most influential nodes in the social network. The problem is classically carried out using classical performance metrics such as accuracy or recall, which is not the end goal of the influence maximization problem. Our work presents an end-to-end learning model, SGREEDYNN, for the selection of the most influential nodes in a social network, given a history of information diffusion. In addition, this work proposes data visualization techniques to interpret the augmenting performances of our method compared to classical training. The results of this method are confirmed by visualizing the final influence of the selected nodes on network instances with edge bundling techniques. Edge bundling is a visual aggregation technique that makes patterns emerge. It has been shown to be an interesting asset for decision-making. By using edge bundling, we observe that our method chooses more diverse and high-degree nodes compared to the classical training.

**Keywords:** influence maximization; end-to-end learning; decision-focused learning; graph visualization; edge bundling; differentiable greedy

## 1. Introduction

The rapid growth of social networks in recent years has sparked extensive research on understanding the dynamics of these networks. The diffusion of information within social networks has significant implications in various fields, including marketing [1], politics [2], and surveillance [3]. Social networks have emerged as powerful platforms for mass information diffusion, influencing major events such as elections and social movements such as the Arab Spring. Exploring the mechanisms of information diffusion is crucial for detecting manipulation attempts, mitigating terrorist risks, and optimizing product advertising.

The problem of information diffusion centers around how information spreads and propagates among users. One fundamental problem in this domain is the Influence Maximization problem, which involves identifying a set of users that maximizes the spread of information. However, this problem is known to be NP-hard, posing computational challenges for classical statistical approaches to studying information cascades [4].

Due to the sudden growth of social networks, the quantity of data to process became unmanageable for classical statistical studies of information diffusion instances, called

information 'cascades'. However, due to the rapid development of these networks, the quantity of data to process became a challenge to handle. In recent years, machine learning models have shown promise in directly estimating the influence that users exert on each other within social networks. By leveraging these predictions, it becomes possible to identify the most influential users in the network. Traditionally, these two steps of prediction and influence maximization are performed sequentially. However, recent research has suggested the joint execution of these steps for improved influence maximization [5].

In this paper, we propose an end-to-end learning approach using machine learning models to predict diffusion probabilities in social networks. Additionally, we developed a novel visualization method to evaluate the quality of our solution compared to existing methods. We argue that visualizing the graph provides deeper insights into diffusion mechanisms than conventional numerical metrics. Building on the concept highlighted in Anscombe's work [6], where statistics fail to capture the full patterns of data, social networks exhibit a similar phenomenon. Applying edge bundling techniques to visualize dense and cluttered graphs allows us to better understand edge connections.

The contributions of this paper are twofold. Firstly, we introduce a novel graph optimization algorithm for maximizing influence in social networks. Secondly, we employ visualization techniques to validate our model's performance against two baseline models. The visualization model enhances our understanding and evaluation of the proposed solution.

The remainder of this paper is organized as follows. Section 1 provides the background and context, while Section 2 reviews and summarizes related work on the influence maximization problem. In Section 3, we describe the data and methods employed in this study. Section 4 presents the results and findings, and Section 5 concludes the paper by summarizing the key findings and discussing future research opportunities.

## 2. Related Work

In this section, we first review the influence maximization problem and existing end-to-end learning methods. Next, we review existing visual simplification techniques for dense data visualization.

### 2.1. Influence Maximization

The study of information diffusion in social networks began in the early 2000s with the seminal work of Kempe et al. [4], in which they propose a greedy framework to find approximations of the optimal subset of nodes maximizing influence with theoretical guarantees. The recent advances in the optimization of submodular functions allowed the improvement of greedy algorithms in a Cost-Effective Lazy Forward algorithm (CELF) [7] and then CELF++ [8], which are much faster.

However, these algorithms require knowing the diffusion probabilities between users, which is problematic on real social networks since this information is not available. To solve this problem, machine learning algorithms have recently been used to learn the influence of content on social networks, forecast the future bursts of popularity of content, or generate new cascades. Decision-trees-based models, support vector machines, and clustering algorithms have been used for a decade to predict the influence of content on social networks. However, since DeepCas [9], they have been progressively replaced by deep learning models.

### 2.2. End-to-End Learning

Recently, a new method for training machine learning models arose for solving complex data pipeline problems. End-to-end learning can be used on "prediction-optimization" problems, where the prediction of a model is then used to optimize a certain quantity. The "prediction-optimization" problems are classically solved in two stages. In the first stage, the model is trained to maximize its accuracy. The outputs of the model tend to be close to the historical data. In the second stage, an optimization algorithm is executed on the predicted values of the model. End-to-end learning differs in that the model is not trained

to maximize the accuracy but is directly optimized to maximize the final influence of the optimal solution found using the predicted probabilities. This framework has recently been applied to recommendation systems [5] but has never been applied to influence maximization on social networks.

## 2.3. Edge-Bundling

Several initiatives related to the visualization of social networks have been developed in recent years; however, there has been limited focus on influence maximization or information cascades [10]. The large quantity of data involved in information diffusion in social networks makes visualization both important and challenging to develop.

Due to the large number of edges and the high density of edges in the graph, edge-bundling techniques can be used to facilitate the interpretation of results. Edge bundling techniques have in common to cluster close edge paths together, thus increasing the number of white spaces and reducing the clutter in layouts of large graphs. Bundling can be seen as sharpening the edge density in the layout, making areas of high density even denser and areas with a lesser edge density appear sparser or white.

As opposed to graph simplification techniques where edges considered unimportant are simply removed from the layout, no edges are removed during edge bundling, and the overall topology of the graph is conserved.

Edge bundling can be used to identify the links between groups of nodes that would be invisible in a large, dense graph due to the clutter of edges. The identification of clusters is made easier by the white spaces separating the clusters. Edge bundling, however, does not conserve the direction of the edges. In certain cases, the direction of the edges can be important, such as in trajectories or geographical data. The direction of the edge is reduced to a small set of main directions, thus losing the initial directions information.

Recent work has tried to conserve the initial edge directions for automobile traffic and airplane trajectories [11]. In our case, the direction of the edges is not important.

Since 2006, with the seminal work of Gasner and Koren, various edge bundling techniques have been developed. Initially, edge bundles were drawn as straight lines based on spatial proximity [12]. Qu et al. added NURBS splines to replace the original straight-line bundles [13]. Most notably, hierarchical edge bundling was developed by Holten [14] to bundle large graphs of several thousands of nodes easily. After this, variants spurred, adapted to whether the graph is static or dynamic, directed or undirected, 3D or 2D [15].

## 3. Data and Methods

In this section, we briefly present the Weibo dataset used to train the models. Furthermore, we detail the specificity of the end-to-end approach we used to train the SGREE-DYNN model.

## 3.1. Dataset

Our approach uses data scraped from real social networks in order to predict the best influencers. The dataset used is scraped from the Chinese micro-blogging social media Weibo [16]. It contains examples of information cascades stored as lists of reposts. Information about user profiles, topic classification of the messages, and the social graph are also available.

Table 1 above provides a summary of our dataset. From this dataset, we extract and use 24 features describing the link $(u, v)$. These features are extracted from the users' profiles, the topology of the social graph, the topic modeling of the posts of the users, and the cascade information. The detailed list of the features is given in Appendix A.

**Table 1.** Dataset summary.

| Dataset | # Cascades | # Users | # Reposts |
|---|---|---|---|
| Weibo | 300 K | 1.7 M | 200 M |

The goal is then to use the end-to-end learning method to train machine learning models to predict the diffusion probabilities between influencers and targets. The ground truth diffusion probabilities used are extracted from the previous examples of information diffusion cascades.

Figure 1 above provides the end-to-end process for the learning framework for maximizing the influence. The social network and its content are preprocessed to create an instance X containing the features vectors $X[u, v]$ of all the pairs of nodes $(u, v)$ and a matrix P containing the ground-truth diffusion probabilities. The instance X is fed to our SGREEDYNN model. This model predicts a diffusion probability matrix, which is then fed to the optimization algorithm. Given these diffusion probabilities, the algorithm chooses the best subset of k influencers among the users in the social network that maximizes the information propagation. The influence function $\sigma$ then returns the real influence of this subset. The weights of the model are then updated to maximize this final influence.

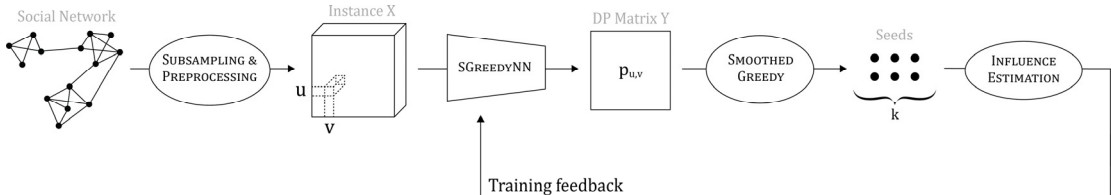

**Figure 1.** General pipeline of the learning framework.

Given a history of actions $\{(u, v)\}$ where $u$ influences $v$, the ground truth diffusion probability $p_{u,v}$ is defined as

$$p_{u,v} = \frac{A_{u,v}}{A_{u,\cdot} + A_{\cdot,v}} \tag{1}$$

where $A_{u,v}$ is the number of times $u$ influenced $v$, $A_{u,\cdot}$ is the number of posts posted by $u$, and $A_{\cdot,v}$ is the total number of reactions of $v$ [17].

The obtained diffusion probabilities are, in reality, very small. The estimated ground truth probabilities are mapped to higher values to facilitate the training. For positive probabilities, the first two deciles are mapped to a probability of 1, the next 3 deciles are mapped to a value of 0.5, the next 3 deciles are mapped to 0.2, and the rest is mapped to 0.1. The probabilities equal to 0 do not change. We chose these values.

To increase the number of positive examples in the diffusion probability matrix during the training, only the targets participating in more than 150 cascades are considered, and only the top 20% of influencers on this induced graph are considered. This gives a subset of influencers $I \subset V$ and $T \subset V$.

Given this sub-sample, $I, T$, the training dataset D is created by randomly drawing $n$ influencers in $I$ and $m$ targets in $T$ and creating a matrix X of size $(n, m, 24)$ where $X[u, v, :]$ is the feature vector of size 24 associated with the potential influence link $(u, v)$. At the same time, a matrix $Y$ of size $(n, m)$ is created with $Y[u, v] = p_{u,v}$ where $p_{u,v}$ has been previously defined. An example of a $Y$ matrix is provided in Appendix A.

### 3.2. End to End Learning

The end-to-end machine learning model is trained on the dataset D = $\{(X, Y)\}$. The model is then optimized to minimize the following function by stochastic gradient descent.

The use of the Smoothed Greedy algorithm (SGREEDY) as an optimization algorithm allows an easy estimation of the objective function's gradient.

$$J(\theta) = - \sum_{(X,Y)\in D} \sigma(\text{SGREEDY}(m(X;\Theta)), Y) \tag{2}$$

The model $m$ is parameterized by $\Theta$, takes an instance $X$ of size $(n, m, 24)$ as an input, and returns a diffusion probability matrix of size $(n, m)$. The Smoothed Greedy algorithm takes a diffusion probability matrix as an input and returns the index of the best seeds in the network. The influence spread function $\sigma$ takes a seed set and a diffusion probability matrix and returns a positive number. Here,

$$\sigma(S) = \sum_{v=1}^{n} \left( 1 - \prod_{u\in S}(1 - p_{u,v}) \right) \tag{3}$$

The model has been trained on 50 instances of size $500 \times 500$, on 100 epochs, with a batch size of 4, a learning rate $\lambda = 5e^{-4}$. The temperature of the Smoothed Greedy algorithm is $\epsilon = 0.1$, and the sample size of the Smoothed Greedy algorithm is 20. The hyperparameters were chosen to maximize the number of targets influenced in the test dataset using the Bayesian optimization and hyperband method [18].

### 3.3. Visual Aggregation

In addition to the other performance metrics, we explain and compare the results found by the models using edge bundling techniques on the different networks studied. These visualizations give new insights into the behavior of our method and how it surpasses the other methods.

The edge bundling algorithm used is the kernel-based estimation edge bundling algorithm [19]. The first step of the algorithm is to estimate the edge density map using the kernel density estimation. Then, the normalized gradient direction is estimated, and the edges are moved in the gradient direction and smoothed by using Laplacian filtering. These steps are repeated with a decreasing kernel size until the result is convincing. Tuning these parameters typically involves a combination of manual experimentation and automated optimization. The following parameters are chosen such that only a few main edge bundles appear while retaining most of the information of individual edges:

- number $n$ of iterations of the algorithm
- tension $t$ of the edges
- initial bandwidth $bw$ of the kernel
- decay rate $d$ of the kernel's size

We apply this technique to an instance from the training dataset. The instance is a graph containing 500 influencers and 500 targets. The graph contains the edge $(u, v)$ if $v$ is present in a cascade initiated by $u$. We call this graph the cascade graph. This graph is bipartite, and the only possible information diffusion is between an influencer and a target. The dense instances contain tens of thousands of edges; thus, the use of edge bundling on this graph is appropriate.

### 3.4. Choice of Graph Layouts

Edge bundling techniques are very dependent on the layout used. We investigated several layouts, as shown in Figure 2 below. We color the edges according to the position of the origin node to better see the direction of the bundled edges. In circular layouts, the color of the edge depends on the angular coordinate of the origin node. In lateral layouts, the color depends on the y-component of the origin node.

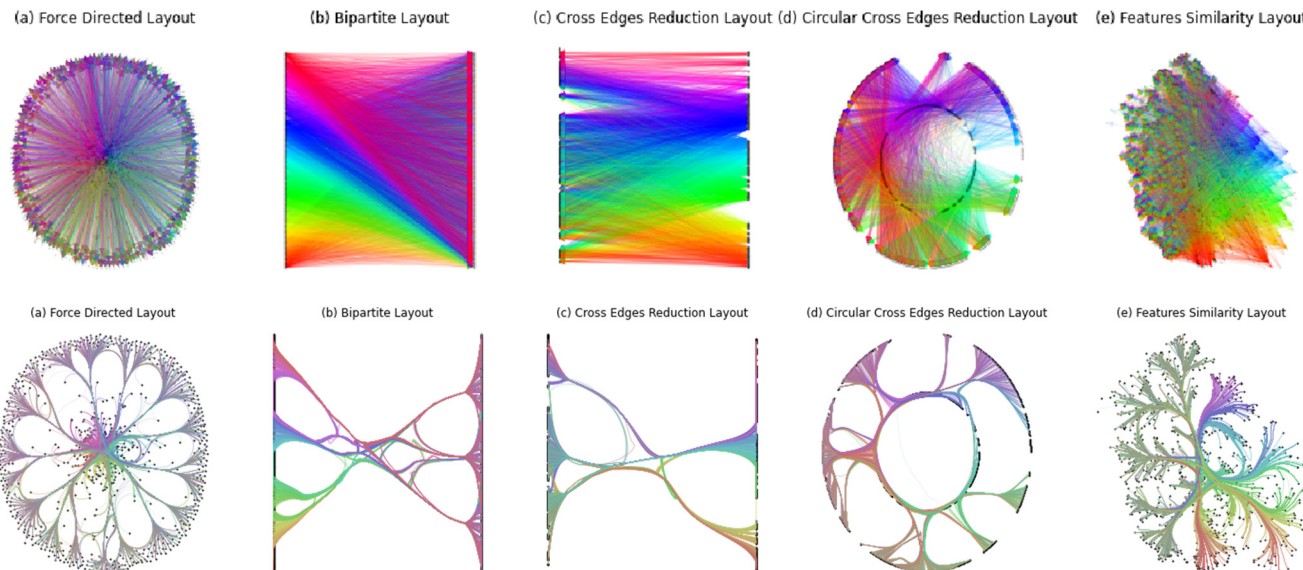

**Figure 2.** Comparison of 5 different layouts in the bundled and unbundled case. Edge bundling is applied on the second row. The five different layouts are, from left to right, (**a**) force-directed, (**b**) bipartite, (**c**) cross-edge reduced bipartite, (**d**) circular cross-edge reduced bipartite, (**e**) similarity. The unbundled graphs are cluttered, which makes any interpretation very difficult. Edge bundling makes the visualization clearer.

Several layouts have been explored. We explain briefly their specificity.

### 3.4.1. Force-Directed Layout

The principle of the force-directed layout is to consider the edges as strings and to minimize the potential energy of the system [20]. The forces acting on the edges tend to group the nodes in clusters. In our case, the nodes having a higher degree are in the middle; they link the peripheral nodes having a lesser degree. By applying edge bundling on this layout and coloring the edges according to the directions, we can notice that the edges arriving in the outer layer of the figure are colored grey. The outer layer mainly consists of the targets, and the grey color comes from all the colors mixing together. This means that the influencers do not spread their influence in a particular direction, and all targets can receive influence from the targets of any color. This layout may thus not be suited to give significant insights when bundled.

### 3.4.2. Bipartite Layout

We also take advantage of the fact that the graph is bipartite by considering a bipartite layout. The bipartite layout consists of displaying the two groups of nodes (influencers and targets) on two parallel lines.

The order of the nodes is important, and the initial order does not give significant insights.

### 3.4.3. Cross Edge Reduced Bipartite Layout

To counter the issue mentioned above, we apply a cross-edge reduction algorithm [21]. As shown in Figure 2c, the edge bundling technique displays bundles of the same color, which is consistent with the behavior of the edge cross-reduction algorithm.

### 3.4.4. Circular Cross-Edge Reduced Bipartite Layout

To better see the edges between influencers and targets, we spread the targets around the influencers and then organized these two in a circular layout. The influencers are positioned in the inner circle, and the targets are in the outer circle.

However, the placement of the nodes does not depend on the features the model used to predict the diffusion probability. This placement only depends on the original cascade graph topology, which is not accessible to the network.

### 3.4.5. Similarity Layout

This observation motivates us to consider a layout where two nodes are close if they have similar behavior. The principle is to generate an undirected weighted graph having the same nodes as the social network. The edges' weights are defined as the cosine similarity between the feature vectors of the two nodes. This dense graph is then pruned by removing the edges having a weight less than a certain threshold, and we then consider its force-directed layout.

In these conditions, two nodes having a very similar feature vector will be close in this layout. The bundling on this graph shows that the graph is clustered into two separate groups, i.e., the influencers and the targets, even if this separation is not performed in the features. This first observation shows that the features of the influencers and the targets are significantly different. To visualize the effect of the features on the layout, we display the values of the features according to the position of the nodes in the layout.

The following observations can be made. The difference between influencers and targets in a social network mainly comes from the number of followers, the number of cascades initiated, and the PageRank metric. The number of friends is higher for targets than for influencers. This may be due to the subsampling of the targets $T$ explained in the methodology. The number of likes reaches its highest value in the boundary between the influencers and the targets.

### 3.4.6. Influence Maximization

The problem of influence maximization consists of finding the subset of nodes maximizing the influence on the social network's population. To do that, we can apply brushing on the selected nodes by the algorithm. If an algorithm returns a seed set $S$, it is possible to see the influence coverage on the edge bundling graph by only plotting the influence edges $(u, v)$ if $u \in S$.

An example of such brushing is shown in Figure 3 below. This figure shows the influence of the number of seeds on the information diffusion. The seeds are selected by the Oracle Greedy algorithm, detailed in the next section. Logically, by adding more seeds to the seed set, the number of nodes reached by the seed set increases. This can be seen by the increasing number of plotted edges in the bundled case.

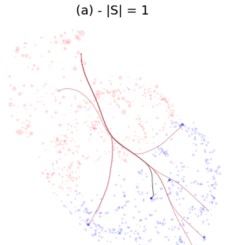 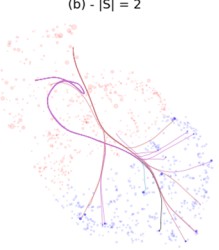 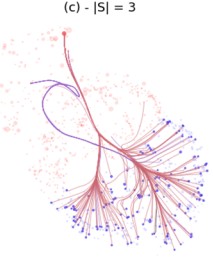 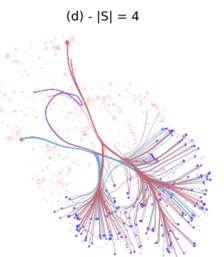 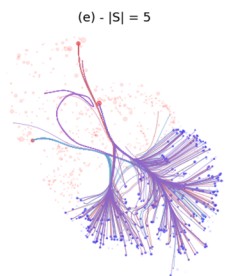

**Figure 3.** Influence of the number of seeds selected on the targets influenced. The red nodes are the influencers, and the blue nodes are the targets. The size of the nodes corresponds to the degree of the nodes in the cascade graph. The color of the edges depends on the position of the origin node, as in Figure 2. The different columns correspond to different numbers of selected seeds $|S|$. The layout is a similarity layout with a weight_threshold of 0.8, an inter-node distance of $k = 0.04$, and 50 iterations. The spread weakly increases when $|S| < 3$ before suddenly increasing with the addition of important nodes to the seed set.

The first observation that can be made on all layouts is the direction of the spread on the first seeds. What can be seen on the edge bundling graph is that the first seeds already

influence the target through all the largest bundles of the diagram. The seed set reaches targets in all directions.

The second observation that can be made thanks to the edge bundling visualization is the difference in spread between the added seeds. In Figure 3b, we can see that the added node does not participate much in the increase of the number of influenced nodes.

The following node added in Figure 3c spreads its influence in all branches and adds more nodes.

## 4. Results & Findings

### 4.1. Introduction

Different layouts have been explored. Due to the weaknesses of the first four layouts, we developed a layout based on the similarity between the feature vectors of the nodes. Figure 2 compares the graph visualization with and without edge bundling. Acquiring insights on the graph topology on the visualization without edge bundling is impossible. The edge bundling techniques give some insights into the 3 cases.

The graph layouts in the unbundled cases are cluttered, which makes any interpretation impossible. The edge bundling technique applied to the different layouts gives clearer visualizations of the relations between the nodes.

### 4.2. Comparison between Algorithms

We compare the results of the three following algorithms. Selecting the best-performing nodes can be separated into two different tasks. An estimation task and an optimization task.

SGREEDYNN: This is our proposed method. This model is trained in the manner explained in Figure 1. The model then infers the diffusion probability matrix of the network.

2STAGE: To prove the performances of the end-to-end learning on influence maximization, we train this model in a classical way. This model is trained to minimize the mean square loss between the predicted and the ground-truth diffusion probability matrices.

ORACLEGREEDY: We also evaluate how well the two estimated matrices match up with the actual diffusion probabilities. These actual probabilities are detailed in Equation (1). To compare the performances, we directly execute the greedy algorithm on the ground-truth diffusion probability matrix.

The optimization part of the task is the same in the three methods. The $k$ nodes constituting the seed set $S$ are selected by a greedy algorithm. Thus, the difference in the three methods is only in how to estimate the diffusion probability matrices.

We compare this visualization using edge bundling on the similarity layout in Figure 4 below. The edge bundling helps to determine which nodes the influencers are targeting. The edge bundling shows that the first influencers to be added are targeting in every direction. However, when the number of seeds increases, the behavior of our method varies from the classical greedy algorithm. While the greedy algorithm continues to add nodes influencing in every direction, the additional seeds added by our model seem to focus on areas where the first seeds had low coverage. The difference between our method and the two other models is visible in Figure 5 below.

For $|S| = 3$, our method chooses three seeds having a high degree and far away on the layout from each other (Figure 4b). The distance between the influencers informs us that our model chooses diverse types of influencer profiles. The number of targets influenced by the seed set is thus increased. Indeed, similar influencers may influence the same targets; thus, different influencers have different sets of targets. This observation contrasts with Figure 5a (OracleGreedy), where the selected nodes are small and close to each other, creating an overlap in the influenced nodes.

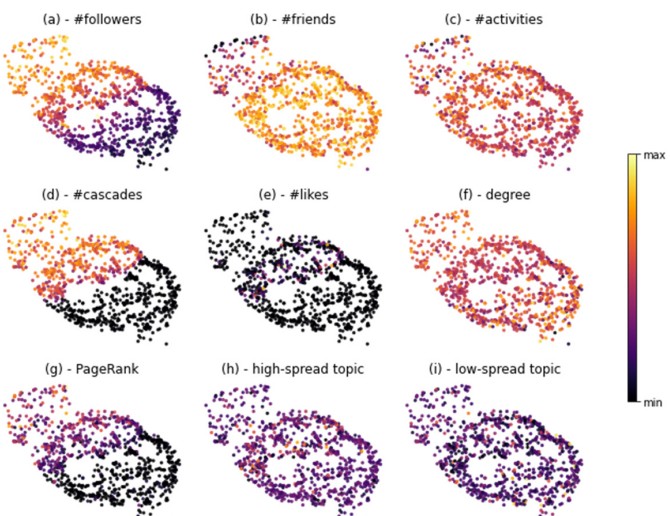

**Figure 4.** Distribution of nine different features on the similarity layout. For each figure, the brightest color corresponds to the highest value, and the darkest color corresponds to the lowest value. The layout used is the same as in Figure 3.

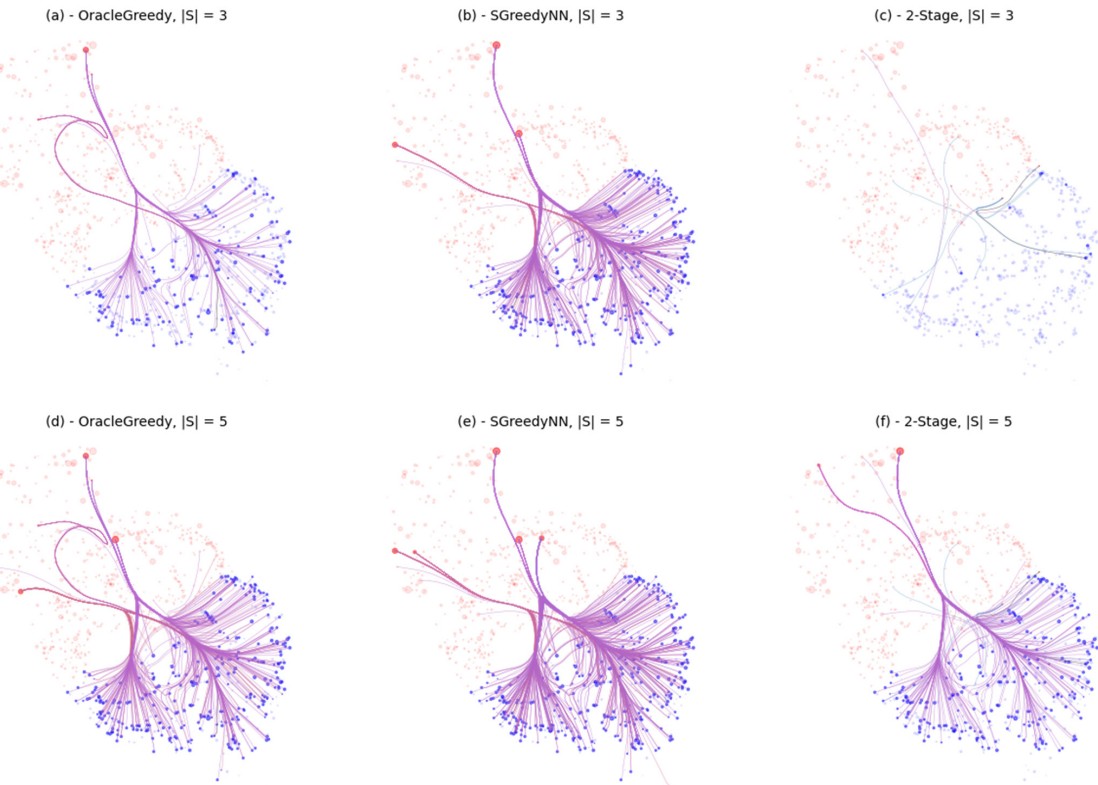

**Figure 5.** Comparison between the three methods. The layout used is the same similarity layout used in Figure 3. The three columns correspond to the three methods compared, and the two rows correspond to two different values of initial number of seeds. They are the Oracle Greedy algorithm in (**a**), our method SGREEDYNN in (**b**), and the classical 2-staged training method in (**c**). The red nodes are the influencers, and the blue nodes are the targets. The size of the nodes correspond to the degree of the nodes in the cascade graph. The color of the edges depends on the position of the origin node. The two rows of the figure correspond to two different numbers of seeds selected. We can see that the performances can greatly vary among the models. With one seed, the decision-focused model already reaches a high number of targets, whereas the 2-Staged and the Oracle greedy algorithms mainly choose small peripheral nodes.

On the next line, corresponding to $|S| = 5$, the differences between the performances of the algorithm are less visible. The gap of influenced targets thus rapidly decreases.

The visualization techniques confirm the superiority of the SGREEDYNN model compared to Oracle Greedy and the 2-staged classical learning. When the number of seeds is very low, the edge bundling helps visualize the direction in which the seeds influence the targets.

## 5. Conclusions & Future Work

In this study, we implemented an end-to-end method to learn the information diffusion probabilities between users in a social network. Our model SGREEDYNN performs better than the classical learning method. In addition, we developed visualization methods to better compare and understand the influence of the social network. The performances of our models have been confirmed by the visualization of the social network using edge bundling.

The method presented here has different advantages compared to the numerical performance metrics normally used in influence maximization. After a high-level analysis of different layouts, we showed that edge bundling techniques applied to the training dataset validated our method compared to the classical training.

Several limitations have been noted and could be investigated in future works. The model can be tested with different types of data. In this work, the preprocessing of the social network's data modified the original topology by subsampling the influencers and targets available. It may be interesting to test the visualization techniques on data subsampled differently. In addition, similar social networks, such as Twitter for example, can also be investigated to verify the results.

Different architectures of the model can be investigated. In this work, we only considered artificial neural networks, but this choice can be extended to different types of machine learning models.

Concerning the visualization of the instances, interactivity could be added to facilitate the brushing and the exploration of the graph.

**Author Contributions:** Conceptualisation: C.H.; Methodology: B.C.; Software: M.M.; Formal Analysis: M.M.; Resources: B.C.; Writing original draft preparation: M.M.; Writing—review and editing: C.A.H., C.H. and B.Y.L.; Visualization: M.M.; Supervision: C.A.H., C.H., B.Y.L. and J.N.S.L.; Project Administration: C.A.H. All authors have read and agreed to the published version of the manuscript.

**Funding:** This research was funded by the National Research Foundation, Prime Minister's Office, Singapore, under its Campus for Research Excellence and Technological Enterprise (CREATE).

**Data Availability Statement:** Data can be provided upon request.

**Acknowledgments:** This research project was conducted during an internship at CNRS@CREATE. I would like to acknowledge their support and provision of necessary resources.

**Conflicts of Interest:** The authors declare no conflict of interest.

## Appendix A. Features Used by the Models

### Appendix A.1. List of Features

To estimate the information diffusion probabilities, our SGREEDYNN model uses the following features. The first column contains $u$ if the feature comes from the influencer, $v$ if it comes from the target, and $u, v$ if it depends on both the influencer and the target. "#" represents, "the number of".

### Appendix A.2. Example of Y

Here is an example of a $Y$ matrix containing the diffusion probabilities $p_{u,v}$ between influencers $u$ and targets $v$ for a subsampling of 500 influencers and targets.

**Table A1.** Features used for estimating the diffusion probability between two users ($u$, $v$).

| User | Feature |
|------|---------|
| $u$ | # followers |
| $u$ | # friends |
| $u$ | # activities |
| $u$ | verified |
| $u$ | gender |
| $u$ | # cascades |
| $u$ | # likes |
| $u$ | # reposts |
| $u$ | out-degree |
| $u$ | PageRank |
| $u$ | high-spread topic |
| $u$ | med-spread topic |
| $u$ | low-spread topic |
| $v$ | # followers |
| $v$ | # friends |
| $v$ | # reposts |
| $v$ | verified |
| $v$ | gender |
| $v$ | in-degree |
| $v$ | PageRank |
| $v$ | high-spread topic |
| $v$ | med-spread topic |
| $v$ | low-spread topic |
| $u,v$ | social edge |

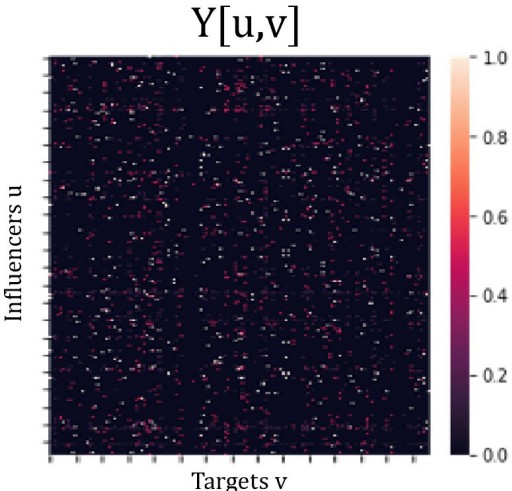

**Figure A1.** Example of a $Y$ matrix used for visualization in this paper.

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
