# Peer review of "Visual Explanations of Differentiable Greedy Model Predictions on the Influence Maximization Problem"

_2504-2289, doi:10.3390/bdcc7030149_

Round 1
Reviewer 1 Report
This work proposes an end-to-end learning model, SGREEDYNN, for selecting the most influential nodes in a social network with a given history of information diffusion.
It is suggested that the introduction should more clearly describe the problems with the present study, that is, why the authors are doing this study and what the study can solve that cannot be solved at present.
It is also suggested that the structure of the article be adjusted: the methods sections of Chapters 3 and 4 may be considered to be merged into "Data and methods". Add a new chapter dedicated to presenting and analyzing the results.
It is also suggested that this article give a technical roadmap to give the reader an idea of what the overall research process looks like.
L130:Could the authors give an example of the data so that the reader can understand what the data looks like?
Figure 1 is so blurry that we can barely make out any text.
It is recommended to enlarge the size of Figures 2-5.
Author Response
Date: 9 Aug 2023
Journal: MDPI
Dear Reviewer
On behalf of myself and all the co-authors, I would like to re-submit our revised manuscript, entitled ‘Visual Explanations of Differentiable Greedy Model Predictions on the Influence Maximization Problem’
as an original research article for publication in the MDPI Journal.
We would like to thank you for your careful assessment and useful suggestions and opportunity to resubmit a revised form of our manuscript. Your comments have been valuable in helping us revise and improve our manuscript. In addition, your comments have helped highlight the significance of our research. We have included our responses to your comments in this response letter.
We hope that our revised manuscript allays all concerns related to our original manuscript. Changes to the original manuscript are marked with track changes in yellow.
All authors have read and approved the final version of this manuscript for submission. None of the authors have a conflict of interest to disclose. No part of this manuscript has been or will be published or submitted or presented elsewhere before appearing in the MDPI Journal.
Thank you for your consideration and hoping to receive favourable outcome soon.
Yours sincerely,
Dr Carol Anne Hargreaves (MBA, MSC, CFE, PhD)
National University of Singapore, Tahir Foundation Building, 12 Science Drive 2, level 10, Singapore 117549.
Email, carol.hargreaves@nus.edu.sg
Tel:+65-6516-6623

Reviewer 2 Report
This research is interesting, it presents a brief introduction to the research to which numerous contributions have been made in recent years. Artificial intelligence is also a factor that we can see in all the gadgets that surround us and not only that.
I appreciate the fact that the authors designed a "SGREEDYNN" method that they compared with those currently used in research and obtained interesting numerical values. He also proposed interpretation techniques for augmenting performances. The method indicates "Influence Maximization problem" algorithms used in social networks.
They also indicated that the research can be expanded but will do so in other articles.
In order to be published, I think that a revision should be made from the point of view of the format of this article but also from the point of view of grammar. For example: I refer to the figures and from my point of view the explanations are not made in a separate sentence from the title of the figure.
Author Response

(The authors gave the same response as above.)

Reviewer 3 Report
In the article entitled "Visual Explanations of Differentiable Greedy Model Predictions on the Influence Maximization Problem," I furnished my comments and concerns here.
In section 3
1. The authors provide a comprehensive overview of the Weibo dataset and the end-to-end approach to training the SGREEDYNN model. However, including more information about the data collection process would be helpful. How was the dataset scraped from Weibo? Were any specific criteria applied to ensure its representativeness? Providing these details would enhance the transparency and replicability of the research.
2. While this section thoroughly describes mapping estimated ground truth probabilities to higher values, it would be beneficial to elaborate on the reasoning behind this mapping scheme. What are the motivations for assigning 1, 0.5, 0.2, and 0.1 probabilities to different deciles? Additionally, are there any potential limitations or implications of this mapping strategy? Including a discussion on these aspects would strengthen the methodology section.
3. This section provides insightful details regarding the model's training, such as the number of instances, epochs, batch size, learning rate, temperature, and sample size. However, discussing the reasons for selecting these specific values would be a good idea. Why was a batch size of 4 chosen? What were the considerations for setting the learning rate to λ = 5e−4? Explaining these choices would enhance the readers' understanding of the experimental setup and reinforce the credibility of the research.
In subsection 4.1 of section 4
1. The authors provide an interesting approach by using edge bundling techniques to visualize and compare the results of their models. However, including some examples or screenshots of the visualizations generated by the Kernel-Based Estimation Edge Bundling algorithm would be helpful for me. This would allow readers to understand better the insights gained from these visualizations and assess the edge bundling technique's effectiveness in showcasing the proposed method's superiority.
2. While the section briefly mentions the different parameters used in the edge bundling algorithm, providing more details about their selection would be helpful. How were the values of the number of iterations, tension of the edges, initial bandwidth of the kernel, and decay rate of the kernel's size determined? Were there any considerations or experiments conducted to find the optimal values? Providing this information would enhance the reproducibility and applicability of the edge bundling technique to other datasets and research studies.
3. The section describes the application of edge bundling on a bipartite cascade graph with 500 influencers and 500 targets. However, it would be a good idea to discuss the reason for choosing this specific instance from the training dataset for visualization. Were there any specific characteristics or patterns in this graph that make it representative of the overall dataset? Additionally, I would like to know if there are plans to apply the edge bundling technique to other instances or real-world datasets to validate its effectiveness further. Including these aspects would provide a broader context for applying edge bundling in the research.
In subsection 4.2 of section 4
1. The section comprehensively analyzes different graph layouts for edge bundling techniques. However, adding a more detailed evaluation and comparison of the layouts would be helpful to me. For instance, the authors could discuss the advantages and limitations of each layout in terms of clarity, interpretability, and scalability. Additionally, providing quantitative measures or user studies to assess the effectiveness of the edge bundling technique in enhancing the visualizations for each layout would further strengthen the paper's findings.
2. While this section briefly explains the principles and characteristics of each layout, it would be helpful to provide references or additional details on the cross-edge reduction algorithm mentioned in Section 4.2.3. The readers would benefit from a clearer understanding of how this algorithm is applied and its specific impact on the bundled visualization. Moreover, if any considerations or trade-offs are associated with using the cross edge reduction algorithm, it would be valuable to discuss them to provide a more balanced perspective.
3. In Section 4.2.6, the authors conclude by stating that edge bundling techniques provide clearer visualizations of the relations between the nodes. However, it would be very helpful to elaborate on the specific insights gained from the bundled visualizations compared to the unbundled cases for each layout. What are the key patterns, trends, or structures become more evident with edge bundling? Providing concrete examples or case studies to support these claims would enhance the paper's contribution and make it more convincing to the readers.
In subsection 4.3 of section 4
1. The authors mention the Oracle Greedy algorithm but do not explain its functioning. Could the authors provide a brief overview or references for readers who are unfamiliar with this algorithm? Additionally, it would be helpful to clarify whether the Oracle Greedy algorithm is commonly used or widely accepted as a benchmark for influence maximization, as this would provide context for comparing the proposed SGREEDYNN method and the 2-Staged classical learning model.
In subsection 4.4 of section 4
1. The authors compare the results of three algorithms for influence maximization. While the performance differences are discussed, providing additional analysis and interpretation of the observed variations would be beneficial. For instance, what factors contribute to the decision-focused model (SGREEDYNN) reaching many targets with only one seed, while the other models mainly select small peripheral nodes? Are there any trade-offs or implications associated with each approach? Providing insights into the underlying reasons for the performance disparities would enrich the discussion.
2. The visualization techniques applied, such as edge bundling on the similarity layout, provide valuable insights into the behavior of the different algorithms for influence maximization. However, adding a more detailed explanation of the visualization results shown in Figure 5 would be helpful to me. For instance, what specific patterns or characteristics can be observed in the visualizations of the different models? How do these visualizations correlate with the algorithms' performance? Elaborating on these aspects would strengthen the paper's findings and provide a more comprehensive understanding of the influence maximization process.
3. It would be interesting to know if the authors considered any other evaluation metrics or measures to compare the performance of the three algorithms for influence maximization. Are there any benchmarks or standard datasets that could be used for comparison? Additionally, if there are any limitations or potential biases in the evaluation methodology, it would be valuable to address them to ensure the reliability of the findings.
English language correction is only little necessary.
Author Response

(The authors gave the same response as above.)

Reviewer 4 Report
Title: Visual Explanations of Differentiable Greedy Model Predictions on the Influence Maximization Problem
The study of social networks has taken advantage of recent advances in machine learning to process the immense amount of data. Automatic emotional labeling of content on social media was made possible by the recent progress in Natural Language Processing. In this work, we are interested in the Influence Maximization problem, which consists in finding the most influential nodes in the social network. The problem is classically carried out using classical performances metrics such as accuracy and recall, though this is not the end goal of the influence maximization problem
Few comments of this study are presented as
-- Introduction is written so poor, which is needed to improve.
-- Provide the organization of the paper at the end of introduction section
-- Use punctuation marks properly
-- Figure 1 is not visible, it must be improved
-- It is always necessary to used the instance is a graph containing 500 influencers and 500 targets.
-- What is the purpose to take three cases?
-- Provide conclusions in such a way that shows the calculated results.
-- Add few more references related to this study as
* Fractional mayer neuro-swarm heuristic solver for multi-fractional order doubly singular model based on lane–emden equation
* Numerical investigations of a new singular second-order nonlinear coupled functional Lane–Emden model
* An efficient stochastic numerical computing framework for the nonlinear higher order singular models
Author Response

(The authors gave the same response as above.)

Round 2
Reviewer 1 Report
Thanks to the authors for their responses, I have no further comments.
Reviewer 4 Report
There is no problem of English